# Association of Circulating miRNAs from the C19MC Cluster and IGF System with Macrosomia in Women with Gestational Diabetes Mellitus

**DOI:** 10.3390/ijms26178367

**Published:** 2025-08-28

**Authors:** Magalhi Guadalupe Robledo-Clemente, Juan Carlos Silva Godínez, Mary Flor Díaz-Velázquez, Edgar Mendoza Reyes, José Damián Gómez Archilla, Lucia Daniela García Montes, Mauricio Ramiro Cortez Chávez, María Isabel Peña-Cano, Renata Saucedo

**Affiliations:** 1Unidad Médica de Alta Especialidad, Hospital de Gineco-Obstetricia No. 3 “Dr. Víctor Manuel Espinosa de los Reyes Sánchez”, Centro Médico Nacional La Raza, Instituto Mexicano del Seguro Social, México City 02990, Mexico; ginemagalhi@outlook.es (M.G.R.-C.); mary.diaz@imss.gob.mx (M.F.D.-V.); gyn.men@hotmail.com (E.M.R.); gomezarchila@hotmail.com (J.D.G.A.); lucia.daniela.garcia.montes@gmail.com (L.D.G.M.);; 2PPCR Program, ECPE Department, Harvard T.H. Chan School of Public Health, Boston, MA 02115, USA; juan.silva@cch.unam.mx; 3Escuela Nacional Colegio de Ciencias y Humanidades, Universidad Nacional Autónoma de México, Mexico City 04510, Mexico; 4Hospital General “Dra. Matilde Petra Montoya Lafragua”, Instituto de Seguridad y Servicios Sociales para los Trabajadores del Estado, México City 13273, Mexico; 5Hospital de Gineco-Obstetricia 221, Instituto Mexicano del Seguro Social, Toluca 50150, Mexico; isabelpenacano@hotmail.com; 6Unidad de Investigación Médica en Enfermedades Endocrinas, Hospital de Especialidades “Dr. Bernardo Sepúlveda Gutiérrez”, Centro Médico Nacional Siglo XXI, Instituto Mexicano del Seguro Social, México City 06720, Mexico

**Keywords:** gestational diabetes mellitus (GDM), fetal macrosomia, C19MC microRNAs, insulin-like growth factor (IGF) system, IGF-binding proteins (IGFBPs), placental biomarkers, epigenetic regulation

## Abstract

Gestational diabetes mellitus (GDM) increases the risk of fetal overgrowth and macrosomia, yet the molecular mechanisms remain unclear. Emerging evidence implicates primate-specific placental microRNAs (miRNAs) from the C19MC cluster in modulating fetal growth via the insulin-like growth factor (IGF) axis. This study aimed to investigate the expression of circulating C19MC miRNAs in GDM pregnancies and their association with IGF axis biomarkers and birthweight outcomes. In this cross-sectional study, 158 pregnant women were stratified into normoglycemic pregnancies (*n* = 52), GDM with normal birthweight (*n* = 56), and GDM with large-for-gestational-age (LGA) newborns (*n* = 50). Plasma levels of 19 C19MC miRNAs and IGF-related proteins were measured. Associations between miRNAs, IGF axis components, and birthweight were analyzed using linear regression and correlation models adjusted for relevant covariates. Several miRNAs, including miR-516a-5p, miR-518d-3p, miR-521, and miR-525-3p, were differentially expressed in GDM, particularly in LGA cases. Strong correlations were observed, such as that of miR-516a-5p with IGFBP-5 (*r* = 0.705; *p* < 0.001). Inverse associations with birthweight were found for miR-519b-3p, miR-518d-5p, and miR-520a-5p. Circulating C19MC miRNAs are dysregulated in GDM and correlate with IGF signaling and fetal growth, supporting their potential as early biomarkers for macrosomia risk in GDM.

## 1. Introduction

Gestational diabetes mellitus (GDM) is defined as glucose intolerance with onset or first recognition during pregnancy and represents a significant public health concern due to its short- and long-term consequences for both mother and child [1]. GDM arises from a progressive increase in maternal insulin resistance coupled with insufficient pancreatic β-cell compensation, culminating in hyperglycemia [2]. This condition is associated with increased risk of adverse perinatal outcomes, including fetal macrosomia, shoulder dystocia, and an elevated likelihood of cesarean delivery [3]. Furthermore, GDM constitutes a major risk factor for future metabolic disorders in both the mother and offspring, such as type 2 diabetes mellitus and cardiovascular disease [4].

Emerging evidence highlights the critical role of the placenta in the pathophysiological mechanisms underpinning GDM and its associated complications, particularly fetal overgrowth [5]. Observational studies have reported increased fetal and placental weights in GDM pregnancies, with placental weight identified as a potential mediator of the effects of maternal hyperglycemia on neonatal birthweight [6,7,8]. Recent research has increasingly focused on the role of microRNAs (miRNAs) in these processes. miRNAs are small, non-coding RNA molecules that regulate gene expression post-transcriptionally by inhibiting the translation of target messenger RNAs, thereby modulating key cellular functions including proliferation, apoptosis, and differentiation [9].

Among the miRNAs implicated in placental biology and GDM, particular attention has been directed toward the chromosome 19 microRNA cluster (C19MC), a primate-specific miRNA cluster composed of 46 miRNAs located at chromosome 19q13.41. This cluster is subject to paternal genomic imprinting and is predominantly expressed in placental trophoblasts, with detectable levels in maternal circulation throughout pregnancy [10]. Individual miRNAs within this cluster have been associated with GDM and fetal development. For instance, miR-518d has been shown to modulate the expression of peroxisome proliferator-activated receptor-α (*PPAR-α*), thereby influencing metabolic regulation in GDM [11]. Other C19MC miRNAs have been linked to neonatal anthropometric outcomes such as birthweight and head circumference [12]. Notably, in silico analyses using the miRDIP database suggest that several members of the C19MC cluster target components of the insulin-like growth factor (IGF) axis, a key hormonal pathway regulating fetal growth. Therefore, C19MC miRNAs may influence fetal growth by regulating IGF gene expression. Moreover, the placenta releases miRNAs into maternal circulation, allowing placental dysfunction to be reflected in maternal blood—potentially serving as a basis for biomarker development [13].

However, despite growing recognition of the importance of C19MC miRNAs, their role in modulating fetal growth through regulation of the IGF axis remains poorly characterized. Given the centrality of the IGF system in fetal growth and its dysregulation in GDM, as well as the predicted regulatory role of C19MC miRNAs in this pathway, there is a compelling need to investigate their potential contribution to fetal overgrowth.

The insulin-like growth factors (IGF-I and IGF-II) play a crucial role as growth hormones during fetal life and are key components in regulating fetal growth. In the placenta, mTOR responds to many growth-related signals, including amino acids, glucose, oxygen, folate, and growth factors, to regulate trophoblast mitochondrial respiration, nutrient transport, and protein synthesis, thereby influencing fetal growth [14]. Previous studies have established the importance of the IGF axis in fetal overgrowth in diabetic pregnancies, yet the integration of epigenetic regulators such as placenta-derived miRNAs into this model remains understudied. While circulating miRNAs from the C19MC cluster have been linked to fetal growth and metabolic signaling, few studies have examined their concurrent association with IGF axis disruptions in GDM pregnancies. Notably, recent data suggest that altered C19MC expression may coincide with aberrant IGF signaling in macrosomia, potentially serving as an early molecular fingerprint of placental dysfunction [15,16,17].

To date, no study has systematically examined the relationship between C19MC miRNA expression, circulating IGF family components, and birthweight in the context of GDM. Therefore, this study aims to provide the first comprehensive analysis of C19MC miRNA expression patterns, IGF family serum concentrations, and their relationship with birthweight in Mexican women with GDM, offering novel insights into the molecular mechanisms linking placental miRNA regulation with fetal growth in diabetic pregnancies in a Mexican population.

## 2. Results

### 2.1. Patients’ Characteristics

A total of 158 women were enrolled and stratified into three groups: 52 normoglycemic pregnant women, 56 women with GDM who delivered newborns with normal birthweight, and 50 women with GDM who delivered newborns with a birthweight at or above the 90th percentile.

The mean maternal age was 31.0 ± 6.1 years (range: 17–43), mean height was 1.59 ± 0.07 m (range: 1.41–1.76), and pre-pregnancy weight was 75.5 ± 16.2 kg (range: 47–138). The average pre-pregnancy BMI was 29.9 ± 5.8 kg/m^2^, with participants distributed across normal weight (19.0%), overweight (38.6%), and obese (42.4%) categories per WHO criteria.

Regarding medical history, 75.3% reported a family history of type 2 diabetes, 10.8% had a prior diagnosis of polycystic ovary syndrome (PCOS), and 3.2% reported tobacco use during pregnancy.

In terms of sociodemographic characteristics, 39.9% had completed high school and 38.0% held university degrees. Most participants were either married (46.8%) or in domestic partnerships (41.8%), and 43.7% were employed, while 38.0% reported being engaged in household activities. Demographic, obstetric, and neonatal characteristics of the study population, stratified by GDM status and birthweight category, are summarized in Table 1.

Women with GDM received individualized treatment based on glycemic control, comorbidities, and clinical judgment. Glycemic control was monitored through capillary glucose testing at least four times per day (fasting and postprandial). Glycemic targets were based on ADA guidelines: fasting glucose < 95 mg/dL, 1 h postprandial < 140 mg/dL, and 2 h postprandial < 120 mg/dL. While individual adherence data were not available, glucose monitoring protocols were consistently implemented throughout pregnancy to support glycemic control. Treatment included medical nutrition therapy, with metformin initiated in those with suboptimal glycemic control, and insulin introduced when oral agents proved insufficient.

### 2.2. miRNA Expression Analysis

Expression profiles of 19 C19MC miRNAs were evaluated across the three groups. Detailed descriptive statistics for each miRNA, including mean, median, and number of samples below the detection threshold in each group, are provided in Appendix A, and relative expression levels normalized to cel-miR-39 are presented in Appendix A. Eight comparisons showed statistical significance to GDM with normal birthweight (reference group); miR-516a-5p (*p* = 0.0025) and miR-525-3p (*p* < 0.001) were significantly downregulated in normoglycemic controls. In contrast, six miRNAs were significantly upregulated in the GDM-LGA group compared to GDM with normal birthweight: miR-518d-3p (*p* = 0.0015), miR-521 (*p* = 0.0080), miR-524-3p (*p* = 0.0429), miR-519b-3p (*p* = 0.0163), miR-512-3p (*p* = 0.0379), and miR-519c-3p (*p* = 0.0185). No other miRNAs showed statistically significant differences. These findings suggest that C19MC miRNAs are differentially regulated both in normoglycemic pregnancies and in GDM complicated by fetal overgrowth when compared to GDM with normal birthweight, revealing distinct miRNA expression patterns associated with glycemic status and macrosomia risk within the context of gestational diabetes (Figure 1A).

### 2.3. IGF Axis and Insulin Resistance Analysis

Group-based comparisons of circulating growth factors and insulin resistance markers revealed statistically significant differences in IGF-2 (*p* < 0.001), HOMA-IR (*p* < 0.001), insulin (*p* = 0.0025), and glucose concentrations (*p* < 0.001). Although concentrations of IGF-1 (*p* = 0.0995) and IGFBP-6 (*p* = 0.0635) exhibited trends toward group differences, these comparisons did not attain statistical significance. IGFBP-7 levels differed significantly across groups (*p* = 0.0335), whereas PLGF and IGFBP-1 through IGFBP-5 showed no significant differences (Figure 1B).

### 2.4. miRNA and IGF Axis Correlation Analysis

Correlation analyses revealed substantial associations between the C19MC miRNAs and maternal biomarkers in GDM patients, with correlation coefficients ranging from moderate to strong (Figure 2). The strongest correlation was observed between miR-516a-5p and IGFBP-5 in GDM women with normal birthweight (*r* = 0.705, *p* < 0.001), followed by miR-512-3p and IGFBP-2 (*r* = −0.688, *p* < 0.05) in the same group. Among the top correlations, several miRNAs showed strong positive associations with IGFBP-4 and IGFBP-5 (*r* > 0.6, *p* < 0.001), predominantly in the GDM normal birthweight group.

Numerous miRNAs exhibited significant correlations with IGF-related proteins, suggesting potential regulatory links. Multiple miRNAs demonstrated strong associations with various IGFBPs, including miR-518d-5p and miR-520a-5p with IGFBP-4 (*r* = 0.604, *p* < 0.001), and miR-517a-3p with IGFBP-4 (*r* = 0.567, *p* < 0.001), all observed in GDM patients with normal birthweight. miR-516a-5p showed significant associations with IGFBP-3, IGFBP-4, IGFBP-5, and IGFBP-6 across the GDM normal birthweight group, supporting its role in modulating the IGF axis. miR-518f-5p was significantly correlated with IGFBP-3 (*r* = 0.404, *p* = 0.011) in the GDM normal birthweight group and showed a moderate positive correlation with IGF-1 (*r* = 0.316, *p* = 0.039) in the GDM-LGA group. miR-525-3p showed a significant correlation with IGFBP-4 and IGFBP-5, reinforcing its involvement in fetal growth regulation. Other miRNAs with multiple associations included miR-524-3p, miR-518f-5p, and miR-517a-3p, which were significantly correlated with IGFBP-3.

Among the miRNAs found to be differentially expressed across study groups, miR-516a-5p exhibited moderate positive correlations with IGF-1 in the GDM-LGA subgroup, demonstrating group-specific regulatory patterns. The observed correlations between miR-521 and insulin suggest potential regulatory interactions between miRNA expression and maternal metabolic signaling pathways. In the GDM-LGA subgroup, miR-524-5p showed a negative correlation with IGFBP-2 (*r* = −0.403, *p* = 0.012), indicating group-specific regulatory dynamics.

To explore the potential links between maternal glycemic control and molecular markers, we performed correlation analyses between glucose values and both miRNA expression and IGF axis components (Appendix A). Several miRNAs, including miR-516a-5p and miR-521, exhibited significant correlations with maternal glucose and insulin concentrations, supporting a potential role in mediating metabolic adaptations in GDM.

### 2.5. Association of Circulating Biomarkers with Risk of LGA Birth

Several maternal biomarkers demonstrated significant associations with large-for-gestational-age (LGA) birth when compared to women with GDM who delivered full-term normal birthweight newborns. Among the IGF axis components, IGFBP-2 and IGFBP-5 showed significant positive associations with LGA in both unadjusted and adjusted models, with adjusted odds ratios of 2.515 and 2.402, respectively (both *p* < 0.001).

Regarding miRNAs, multiple biomarkers showed significant inverse associations with LGA. In adjusted models, eight miRNAs retained statistical significance: miR-512-3p, miR-516a-5p, miR-518e-5p, miR-518f-5p, miR-519b-3p, miR-519c-3p, miR-521, and miR-523-3p (all *p* < 0.05), suggesting protective effects against LGA birth.

Other biomarkers, including IGF-1, IGF-2, and several additional miRNAs, showed non-significant associations in adjusted models. Complete results for all biomarkers are presented in Table 2.

## 3. Discussion

This study explored whether circulating miRNAs from the C19MC cluster contribute to fetal overgrowth in GDM via dysregulation of the IGF axis. We identified eight miRNAs as being differentially expressed when comparing normoglycemic pregnancies and GDM with LGA newborns to GDM with normal birthweight. Notably, miR-516a-5p and miR-525-3p were downregulated in controls, while miR-518d-3p, miR-521, miR-524-3p, miR-519b-3p, miR-512-3p, and miR-519c-3p were upregulated in GDM-LGA pregnancies. miR-525-3p was significantly downregulated in normoglycemic pregnancies compared to GDM with normal birthweight (*p* < 0.001), while miR-516a-5p was upregulated in the GDM group (*p* = 0.0163) and positively correlated with multiple IGF-binding proteins, supporting a regulatory role in IGF signaling [18]. These findings are consistent with prior reports implicating these miRNAs in placental development and maternal–fetal signaling [19], and may reflect compensatory responses to altered maternal metabolic status in GDM [2].

Mechanistically, the dysregulation of C19MC miRNAs observed in GDM pregnancies, particularly those complicated by macrosomia, may alter placental signaling by modulating the IGF axis. For example, both miR-516a-5p and miR-525-3p were significantly upregulated and exhibited strong associations with IGF-binding proteins, corroborating prior evidence suggesting that C19MC miRNAs contribute to trophoblast proliferation and differentiation through IGF-related pathways [18,20].

Other miRNAs, including miR-518d-3p and miR-521, also displayed group-specific associations with IGFBPs and maternal insulin levels, suggesting their potential involvement in placental responses to metabolic stress in GDM-LGA pregnancies. These patterns reinforce the hypothesis that individual C19MC miRNAs may act as dynamic regulators of maternal–fetal signaling in a context-dependent manner. Together, these observations support a conceptual model in which C19MC miRNAs mediate a multi-layered regulatory network, integrating maternal metabolic cues such as hyperinsulinemia and IGF axis dysregulation that drives altered placental function and fetal growth [20].

Our findings further revealed consistent IGF axis dysregulation in GDM pregnancies, particularly those complicated by macrosomia. Maternal IGF-2 concentrations were significantly elevated in the GDM-LGA group (*p* < 0.001), confirming its established role in promoting fetal overgrowth through mitogenic and anabolic effects [17]. These observations are consistent with prior work showing that IGF-2 promotes placental nutrient transport and cell proliferation, both critical for fetal development in diabetic pregnancies [21]. IGFBP-7 levels were also elevated in this group, supporting its possible role in modulating IGF availability and trophoblast differentiation [22].

Logistic regression analyses further supported these associations. IGFBP-2 and IGFBP-5 emerged as significant independent predictors of LGA birth in adjusted models (OR = 2.515 and 2.402, respectively; *p* < 0.001), consistent with prior studies linking first-trimester IGF-1 and IGF-1/IGFBP-1 ratios to macrosomia [23]. These results strengthen the evidence for the role of IGF-2 as a key driver of fetal overgrowth in GDM, particularly through its influence on placental proliferation and nutrient transport [18]. In addition to IGF axis proteins, miR-512-3p, miR-516a-5p, miR-518e-5p, miR-518f-5p, miR-519b-3p, miR-519c-3p, miR-521, and miR-523-3p were inversely associated with LGA birth in adjusted models, suggesting a functional role in fetal growth modulation rather than a passive biomarker effect. These findings are in line with previous studies implicating miR-519b-3p in trophoblast migration and IGF axis regulation [18].

Together, these results highlight the potential of integrating C19MC miRNAs and IGF axis components into predictive models for macrosomia risk stratification. The observed inverse correlations between birthweight and miRNAs such as miR-519b-3p, miR-518f-5p, and miR-520a-5p (*r* = −0.271 to −0.193, all *p* < 0.05), along with positive associations with IGF-2 and IGFBP-2/5 (*r* = 0.237–0.487, all *p* < 0.005), support a coordinated regulatory framework [24].

IGFBP-7, although less studied than IGFBPs 1–6, has emerged as a context-dependent modulator in pregnancy by influencing IGF activity, cell adhesion, and decidualization [16]. Our findings complement reports indicating that maternal obesity and GDM are associated with altered IGFBP-7 concentrations, potentially reflecting disrupted placental adaptation and insulin sensitivity [16,22]. These overlapping disruptions suggest that C19MC miRNAs and IGF-binding proteins operate within a dysregulated molecular environment in GDM that promotes fetal overgrowth.

Group-specific miRNA–protein correlations, such as miR-524-5p’s inverse association with IGFBP-2 in the GDM-LGA group (*r* = −0.403, *p* = 0.0121), and positive correlations between miR-516a-5p or miR-518f-5p with IGF-1, further support dynamic interactions modulated by maternal metabolic status. These findings are consistent with the placenta-specific and paternally imprinted expression pattern of the C19MC cluster, known to mediate maternal–fetal signaling [10]. The early detection of C19MC miRNAs in maternal plasma from the first trimester positions them as promising biomarkers [25,26]. Furthermore, their predicted regulation of genes such as *IGF1R* and *EED* adds an epigenetic dimension to our model, integrating post-transcriptional regulation with metabolic cues to influence fetal growth trajectories [24,27].

While these findings are promising, our study has some limitations that should be acknowledged. We employed a cross-sectional design, which restricted our ability to infer causality between observed miRNA and IGF changes and birthweight outcomes. Moreover, the study design followed a negative control structure to discern miRNA and IGF axis alterations specific to GDM and those uniquely associated with LGA, although this approach could not establish temporal relationships. Longitudinal studies tracking miRNA expression and IGF axis components throughout pregnancy are essential to establish causal relationships and temporal dynamics, which could inform targeted interventions for at-risk pregnancies. Differences in delivery modes across groups may have introduced bias, particularly as cesarean delivery was more frequent in the GDM-LGA group. Furthermore, we lacked data on several important potential confounders, including detailed maternal nutritional profiles; continuous glycemic control metrics, such as daily glucose monitoring values or HbA1c and information on GDM therapy (e.g., diet, insulin, or oral agents); and genetic predispositions, all of which may influence miRNA expression and fetal growth trajectories independently. Our analysis focused on circulating miRNA expression without examining downstream gene targets within placental tissue, which limited insights into the functional impact of these miRNAs on IGF axis modulation. Although clinically accessible, maternal plasma analysis may incompletely reflect tissue-level miRNA dynamics due to compartment-specific expression and differences in miRNA stability [10,25]. Additionally, the absence of corresponding mRNA transcript analysis for the growth factors evaluated at the protein level (e.g., IGF-1, IGFBPs) precluded the comprehensive assessment of transcriptional regulation mechanisms. Maternal adiposity, highly prevalent in GDM, may also confound miRNA expression, particularly for C19MC members [15]. Additionally, our study did not evaluate the potential influence of genetic background or biological variability among participants. While this limitation may have affected inter-individual expression variability, it was not feasible to assess due to the absence of genotypic or ancestry markers. Finally, our use of RT-qPCR provided relative, rather than absolute, quantification of miRNAs. While sufficient for comparative purposes, future studies using next-generation sequencing could yield a more comprehensive expression profile and detect subtle transcriptomic changes.

Despite these limitations, the clinical implications of our findings are notable, especially in contexts like Mexico, where GDM affects an estimated 10–14% of pregnancies but is often underdiagnosed [28]. Macrosomia significantly increases the risk of maternal and neonatal complications, including birth injuries, cesarean delivery, and future metabolic disorders. Moreover, macrosomic infants born to mothers with GDM may be at higher risk for subsequent childhood obesity and metabolic syndrome, accentuating the long-term impact of prenatal epigenetic dysregulation [29]. High glucose concentrations have been shown to alter DNA methylation at the IGF-2/H19 gene promoters, leading to increased IGF-2 expression, one of the most critical growth-promoting genes during fetal development. This epigenetic regulation may represent a key mechanism linking maternal hyperglycemia to fetal overgrowth [30]. Given the high prevalence of GDM in Mexico, integrating miRNA profiling, particularly of C19MC members like miR-519b-3p, into prenatal screening could enhance early detection and management strategies, potentially reducing the incidence of macrosomia and associated complications. Given the cross-sectional nature of our study and the timing of sample collection at delivery, our findings should be interpreted as hypothesis-generating rather than conclusive evidence of causal or predictive mechanisms.

Future research directions should involve multicenter, prospective studies with larger sample sizes and longitudinal designs to capture the temporal interplay of miRNA expression and fetal growth. The integration of multi-omics approaches, genomics, transcriptomics, proteomics, and metabolomics, will allow for a more comprehensive understanding of the placental regulatory network and its disruption in GDM. Such integrative studies could also investigate how nutrient deficiencies, inflammatory states, or other environmental factors interact with genetic and epigenetic mechanisms to drive GDM-related fetal overgrowth [31,32,33]. Ultimately, these efforts can lead to the identification of novel therapeutic targets and preventive strategies to mitigate adverse pregnancy outcomes associated with GDM.

## 4. Materials and Methods

### 4.1. Design and Study Population

We conducted a cross-sectional study at the Unidad Médica de Alta Especialidad, Hospital de Gineco-Obstetricia No. 3 “Dr. Víctor Manuel Espinosa de los Reyes Sánchez,” Centro Médico Nacional La Raza of the Instituto Mexicano del Seguro Social (IMSS), a major referral center for obstetric patients, providing specialized maternal–fetal care to high-risk pregnancies in Mexico City.

Eligible participants were women aged 18 to 45 years with a singleton pregnancy who were diagnosed with gestational diabetes mellitus (GDM) based on the American Diabetes Association (ADA) criteria. GDM was defined as having one or more of the following glucose values meeting or exceeding the thresholds during a 75 g oral glucose tolerance test (OGTT): fasting plasma glucose ≥ 92 mg/dL, 1 h plasma glucose ≥ 180 mg/dL, or 2 h plasma glucose ≥ 153 mg/dL [34]. Inclusion was limited to women who delivered a full-term newborn between 37 and 40 weeks of gestation, with birthweight at or above the 90th percentile during the third trimester, and those with singleton pregnancies. Participants were recruited from the continuous admission area, surgical recovery, or hospitalization units of the hospital. To minimize selection bias, all eligible women who provided consent were enrolled consecutively throughout the study period. Blood samples were obtained on the day of delivery, either during cesarean section or vaginal birth. As spontaneous labor may influence miRNA levels, this represents a potential source of variability not controlled for in the current design. Two comparison groups were also included: (1) women without GDM who delivered a full-term newborn with normal birthweight, and (2) women with GDM who delivered a full-term newborn with normal birthweight.

Exclusion criteria included known cardiac disease, rheumatologic disorders, endocrine conditions (including Addison’s disease and thyroid dysfunction), and a history of malignancy. Following ADA guidelines, women diagnosed with diabetes in the first trimester were considered to have pregestational type 2 diabetes and were excluded. Patients with any hypertensive disorder and those with fetuses with malformations were also excluded. Participants were removed from the study if they withdrew consent after data collection, either verbally or in writing. Additionally, blood samples were excluded if deemed inadequate for analysis, if participants rescinded consent during the study, or if proper specimen handling (e.g., cold chain maintenance until ultra-freezing or processing) was compromised. Written informed consent was obtained from all participants prior to enrollment.

### 4.2. Sample Collection and Processing

Immediately prior to delivery (elective cesarean section or vaginal birth), peripheral blood samples were obtained from participants via venipuncture of the cubital vein. Maternal venous blood (approximately 10 mL) was collected into EDTA-coated tubes to prevent coagulation. Samples were centrifuged at 3000 rpm for 10 min at room temperature to obtain plasma. Following centrifugation, all samples were immediately aliquoted and stored at −80 °C until processing. RNA extraction was performed on previously frozen plasma samples, which were thawed once for initial analysis. In a subset of samples analyzed across different days for inter-assay precision, additional aliquots were thawed once independently. All samples were non-hemolyzed, and no aliquot was subjected to more than one freeze–thaw cycle. All extractions and downstream procedures were conducted using identical reagent lots, kits, and Real-Time PCR instrumentation to ensure procedural consistency across all samples. Thermo Fisher Scientific kits were used throughout, including extraction, reverse transcription, and qPCR. To reduce technical variability, a single operator (R.S.) performed all pipetting steps for RNA extraction, reverse transcription, and qPCR setup.

#### miRNA Isolation

Total RNA was isolated from 200 mL of plasma using the plasma/serum RNA/DNA purification mini kit (Norgen Biotek Corp, Thorold, ON, Canada) following the manufacturer’s instructions. Synthetic C. elegans microRNA (cel-miR-39, Norgen Biotek Corp) was added during the RNA extraction procedure for subsequent normalization in quantitative Real-Time PCR (RT-qPCR). RNA was quantified using Qubit miRNA assay kits (Qubit^®^, Thermo Fisher Scientific, Waltham, MA, USA).

A total of 4 ng/mL of total RNA was converted into complementary DNA (cDNA) by using TaqMan^®^ Advanced miRNA cDNA Synthesis Kit (Thermo Fisher Scientific) according to the manufacturer’s recommendations.

### 4.3. Quantitative Real-Time PCR (RT-qPCR) Analysis

The cDNA level was quantified by Real-Time PCR on the StepOnePlus™ Real-Time PCR System (Applied Biosystems™, Foster City, CA, USA) using Taqman^®^ Universal PCR Master Mix and the following Taqman^®^ Advanced miRNA Assays (Thermo Fisher Scientific, Waltham, MA, USA): hsa-miR-512-3p (identification number 478971), hsa-miR-512-5p (identification number 478972), hsa-miR-516a-5p (identification number 478978), hsa-miR-516b-5p (identification number 478979), hsa-miR-517a-3p (identification number 479485), hsa-miR-518d-3p (identification number 479393), hsa-miR-518d-5p (identification number 479530), hsa-miR-518f-5p (identification number 479532), hsa-miR-519a-3p (identification number 479534), hsa-miR-519b-3p (identification number 479333), hsa-miR-519c-3p (identification number 479495), hsa-miR-521 (identification number 478149), hsa-miR-523-3p (identification number 478994), hsa-miR-518e-5p (identification number 479491), hsa-miR-524-3p (identification number 479338), hsa-miR-524-5p (identification number 479285), hsa-miR-525-3p (identification number 478995), hsa-miR-525-5p (identification number 479396), and cel-miR-39-3p (identification number 478293). A sample was considered positive if the amplification signal occurred before the 40th threshold cycle. The RT-qPCR was examined for target miRNAs and cel-miR-39 simultaneously. All reactions were run in duplicate. In some cases, inter-assay precision was assessed by replicating the same samples on different days. Consistency across runs confirmed assay reproducibility. No-template controls were included in every qPCR plate to detect potential contamination or primer-dimer artifacts. Prior to analysis, the Real-Time PCR system was calibrated using manufacturer-specified calibration plates to ensure measurement accuracy. Relative miRNA levels were calculated by a comparative threshold cycle CT method (2^−ΔCT^, in which ΔCT = CTsample-CTcel-miR-39). All miRNA assays were obtained from the TaqMan^®^ Advanced miRNA Assays (Thermo Fisher Scientific), which were predesigned and validated by the manufacturer to ensure optimal amplification efficiency, specificity, and minimal cross-reactivity. Probe sequences were proprietary. This study followed the Minimum Information for Publication of Quantitative Real-Time PCR Experiments (MIQE) guidelines to ensure experimental transparency, reproducibility, and data quality [35].

### 4.4. Immunoassays

Circulating human placental growth factor (PLGF) concentration was measured by the enzyme-linked immunosorbent assay technique (MilliporeSigma, St. Louis, MO, USA) according to the manufacturer’s instructions. Circulating levels of human IGF-I, IGF-II, and IGF binding proteins 1–7 were assessed using a multiplex immunoassay based on Magpix technology (MILLIPLEX® MAP; MilliporeSigma, Burlington, MA, USA). Insulin concentrations in plasma were quantified using a chemiluminescent immunoassay on the Alinity Analyzer (Abbott Diagnostics, Abbott Park, IL, USA). All assays were conducted in duplicate, and intra- and inter-assay coefficients of variation (CVs) were maintained within acceptable limits (<10%). Insulin resistance was calculated using the homeostasis model assessment index (HOMA-IR) as [fasting insulin concentration (μU/mL) × fasting glucose concentration (mmol/L)]/22.5.

### 4.5. Statistical Analysis

Descriptive statistics were used to summarize study population characteristics. Continuous variables are reported as means and standard deviations (SDs), while categorical variables are presented as proportions. Missing data were addressed through mean imputation.

For the primary comparison across study groups, each biomarker (miRNAs and growth factors) was modeled as a dependent variable using linear or logistic regression, depending on the distributional characteristics of each outcome. Given that relative miRNA expression (2^−ΔCt^) typically follows a skewed, non-normal distribution, we applied regression for miRNAs to better capture group-level differences in central tendency without relying on assumptions of normality. We did not perform absolute quantification or standard curve-based calibration; all analyses were based on relative expression methods. Growth factor data, which more closely approximated normality after rescaling, were analyzed using linear regression. In all models, study group was the main explanatory variable, and we adjusted for maternal age, body mass index (BMI) category, final gestational weight gain, height, smoking status, history of polycystic ovary syndrome (PCOS), treatment modality, physical activity level, gestational age at delivery, newborn sex, birthweight, birth length, maternal serum glucose, and insulin concentrations. GDM mothers with normal birthweight served as the reference group for all comparisons. Additionally, correlation analyses were performed to investigate relationships between the 20 C19MC miRNAs and maternal growth factors, including placental growth factor (PlGF), insulin-like growth factor binding proteins (IGFBPs 1–7), and insulin-like growth factors (IGF-1, IGF-2).

To evaluate associations between circulating miRNAs, IGF axis biomarkers, and large-for-gestational-age (LGA) birth, logistic regression models were fitted with LGA status as the binary outcome. Both unadjusted and adjusted models were estimated, with adjusted models including maternal age, final gestational weight gain, maternal height, gestational age at delivery, maternal glucose and insulin concentrations, BMI classification, smoking status, polycystic ovary syndrome history, treatment type, physical activity level, newborn sex, and GDM status. Women with GDM who delivered a full-term newborn with normal birthweight served as the reference group for all analyses.

Linear regression analyses only reported adjusted models to focus on covariate-adjusted expression differences across groups, whereas logistic regression included both unadjusted and adjusted analyses to evaluate bivariate associations and identify independent predictors of LGA risk.

Statistical significance was defined as a two-tailed *p*-value < 0.05. To control for multiple testing, false discovery rate (FDR) correction using the Benjamini–Hochberg method was applied. Data management and statistical analyses were performed using R version 4.0 (R Foundation for Statistical Computing, Vienna, Austria) and Stata BE 17.0 (StataCorp LLC, College Station, TX, USA).

## 5. Conclusions

This study identified statistically significant differences in the expression of selected C19MC microRNAs and IGF axis components among women with GDM, particularly those delivering large-for-gestational-age (LGA) newborns. Notably, miR-516a-5p and miR-525-3p were differentially expressed in both GDM groups compared to normoglycemic pregnancies, while other miRNAs, such as miR-519b-3p, miR-521, and miR-524-3p, showed increased expression in GDM-LGA cases relative to GDM with normal birthweight. Several miRNAs also demonstrated group-specific correlations with IGF-binding proteins.

These findings highlight the potential of C19MC miRNAs as molecular mediators of maternal–fetal communication in pregnancies affected by gestational diabetes. Their detectable presence in maternal plasma and their association with both biochemical markers and birthweight outcomes suggest that they may serve as biomarkers for fetal overgrowth risk, providing a foundation for developing early detection strategies through sampling at earlier gestational stages. Further longitudinal studies and mechanistic investigations are needed to clarify the causal pathways involved and to assess the clinical utility of miRNA profiling for risk stratification and targeted intervention in gestational diabetes.

## Figures and Tables

**Figure 1 ijms-26-08367-f001:**
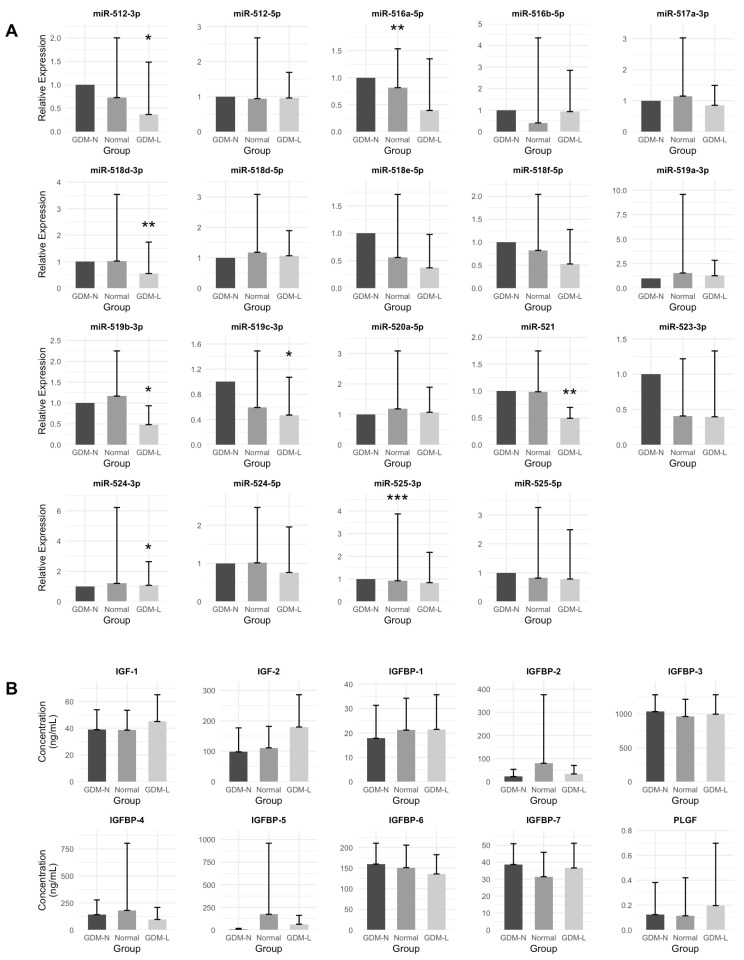
Relative expression of circulating C19MC miRNAs and maternal growth factor concentrations. (**A**) Bar plots depict relative plasma expression of 19 C19MC miRNAs across three groups: Normal (normoglycemic pregnancies with normal birthweight), GDM-N (GDM with normal birthweight), and GDM-L (GDM with LGA newborns, ≥90th percentile). Expression values were normalized to the GDM-N median for each miRNA. Bars represent group medians; error bars indicate the upper interquartile range (75th percentile). (**B**) Bar plots depict mean ± SD concentrations of IGF-I, IGF-II, IGFBP-1 through IGFBP-7, and PLGF across the same three study groups. Concentrations are presented in ng/mL. Asterisks indicate significant differences versus Group 1 in adjusted linear models (* *p*  <  0.05, ** *p*  <  0.01, *** *p* < 0.001).

**Figure 2 ijms-26-08367-f002:**
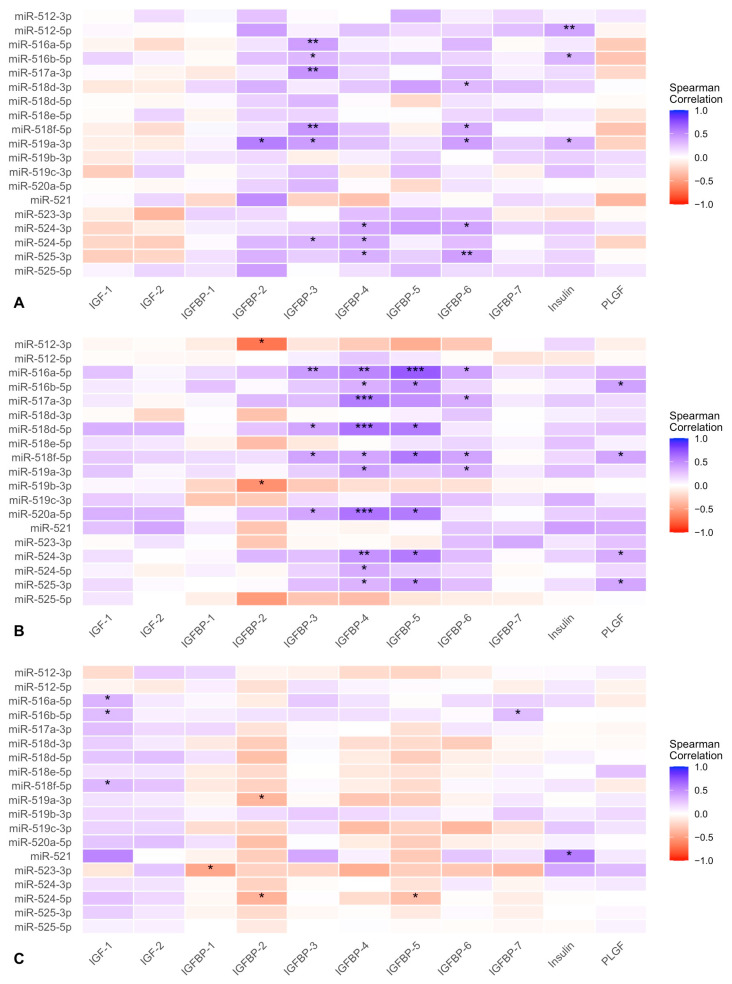
Correlation heatmaps between circulating C19MC miRNAs and maternal insulin-like growth factor axis biomarkers. Spearman correlation coefficients were calculated between 19 C19MC miRNAs and maternal concentrations of placental growth factor, insulin-like growth factors, IGF-binding proteins 1 through 7, and insulin. Heatmaps are shown for each study group: panel (**A**) corresponds to normoglycemic pregnancies with appropriate-for-gestational-age newborns; panel (**B**) includes pregnancies complicated by GDM with appropriate-for-gestational-age newborns; and panel (**C**) represents GDM with LGA newborns. Color intensity indicates the strength and direction of the correlation, with red denoting positive and blue denoting negative associations. Asterisks indicate statistical significance (* *p* < 0.05, ** *p* < 0.01, *** *p* < 0.001).

**Table 1 ijms-26-08367-t001:** Maternal demographic, clinical, and neonatal characteristics of the study populationption.

Characteristic	No GDM, Normal Weight (*n* = 52) ^1^	GDM, Normal Weight (*n* = 56) ^1^	GDM, Weight ≥ 90% (*n* = 50) ^1^	*p*-Value ^2^
Maternal age (years)	29.6 (5.9)	31.8 (5.6)	31.7 (6.7)	0.11
Pre-pregnancy BMI ^3^ (kg/m^2^)	26.9 (4.4)	30.9 (5.7)	31.7 (6.0)	<0.001
Pre-pregnancy weight (kg)	67.1 (12.5)	78.1 (16.2)	81.2 (16.3)	<0.001
Weight in first trimester (kg)	67.6 (12.6)	78.5 (14.8)	82.7 (16.0)	<0.001
Final weight gain (kg)	8.9 (6.7)	7.9 (8.7)	8.8 (8.2)	0.7
Height (m)	1.6 (0.1)	1.6 (0.1)	1.6 (0.1)	0.4
Family history of DM	35 (67%)	46 (82%)	38 (76%)	0.2
Previous PCOS ^4^	4 (7.7%)	7 (13%)	6 (12%)	0.7
Smoking	3 (5.8%)	1 (1.8%)	1 (2.0%)	0.4
Total pregnancies				0.12
1	9 (17%)	18 (32%)	10 (20%)	
2	17 (33%)	15 (27%)	23 (46%)	
3	10 (19%)	14 (25%)	7 (14%)	
4 or more	16 (31%)	9 (16%)	10 (20%)	
Fasting glucose ^5^ (mg/dL)	81.9 (7.1)	102.7 (12.9)	117.3 (19.5)	<0.001
Glucose at 1 h (mg/dL)	127.5 (19.9)	162.1 (22.8)	170.7 (36.9)	<0.001
Glucose at 2 h (mg/dL)	115.6 (18.4)	146.7 (28.9)	161.6 (29.3)	<0.001
Maternal insulin at delivery (uU/mL)	7.7 (8.3)	7.7 (4.8)	11.7 (6.3)	0.002
Maternal HOMA ^6^ index at delivery	1.5 (1.5)	1.9(1.3)	2.8(1.6)	<0.001
Gestational weeks at delivery	38.4 (1.0)	38.5 (0.8)	37.9 (1.2)	0.005
Birthweight (g)	2966.9 (284.1)	3132.3 (298.7)	3988.7 (115.8)	<0.001
Birth length (cm)	49.2 (1.7)	49.8 (1.4)	49.4 (3.4)	0.4
Newborn sex				0.7
Male	27 (53%)	31 (55%)	24 (48%)	
Female	24 (47%)	25 (45%)	26 (52%)	
Delivery mode				0.013
Vaginal	21 (40%)	26 (46%)	10 (20%)	
Cesarean	31 (60%)	30 (54%)	40 (80%)	

^1^ Mean (SD); *n* (%); GDM: gestational diabetes mellitus; ^2^
*p*-values are from one-way analysis of variance (ANOVA) for continuous variables and Pearson’s Chi-squared test for categorical variables; ^3^ BMI: body mass index; ^4^ PCOS: polycystic ovary syndrome; ^5^ glucose values are derived from the OGTT; ^6^ homeostatic model assessment of insulin resistance.

**Table 2 ijms-26-08367-t002:** Association between maternal biomarkers and being large for gestational age: unadjusted and adjusted logistic regression results.

Biomarker	Unadjusted OR (95% CI)	*p*-Value	Adjusted OR (95% CI) ^1^	*p*-Value
Growth Factors				
IGF-1	1.591 (0.715–3.543)	0.255	1.016 (0.365–2.830)	0.975
IGF-2	1.394 (1.066–1.822)	0.015 *	1.250 (0.935–1.670)	0.132
IGFBP-1	1.484 (0.915–2.407)	0.109	1.837 (0.944–3.576)	0.073
IGFBP-2	2.505 (1.617–3.880)	<0.001 *	2.515 (1.468–4.308)	<0.001 *
IGFBP-3	0.523 (0.130–2.103)	0.361	0.474 (0.075–2.998)	0.428
IGFBP-4	0.881 (0.697–1.115)	0.294	0.904 (0.682–1.197)	0.480
IGFBP-5	2.219 (1.641–3.000)	<0.001 *	2.402 (1.582–3.645)	<0.001 *
IGFBP-6	0.256 (0.081–0.809)	0.020 *	0.306 (0.077–1.221)	0.093
IGFBP-7	0.614 (0.219–1.722)	0.354	0.536 (0.146–1.972)	0.348
PLGF	1.002 (0.799–1.255)	0.989	0.960 (0.729–1.264)	0.772
Insulin	3.289 (1.661–6.513)	<0.001 *	1.836 (0.212–15.927)	0.581
miRNAs				
miR-512-3p	0.751 (0.619–0.912)	0.004 *	0.695 (0.545–0.888)	0.004 *
miR-512-5p	0.945 (0.752–1.187)	0.624	0.900 (0.677–1.196)	0.468
miR-516a-5p	0.741 (0.600–0.916)	0.006 *	0.669 (0.509–0.880)	0.004 *
miR-516b-5p	0.987 (0.828–1.177)	0.885	0.993 (0.785–1.256)	0.952
miR-517a-3p	0.882 (0.680–1.144)	0.344	0.798 (0.573–1.111)	0.182
miR-518d-3p	0.830 (0.705–0.976)	0.024 *	0.903 (0.752–1.083)	0.270
miR-518d-5p	1.009 (0.783–1.300)	0.943	0.944 (0.694–1.284)	0.714
miR-518e-5p	0.779 (0.644–0.942)	0.010 *	0.739 (0.572–0.954)	0.021 *
miR-518f-5p	0.739 (0.609–0.896)	0.002 *	0.749 (0.594–0.944)	0.014 *
miR-519a-3p	0.971 (0.820–1.150)	0.734	0.889 (0.707–1.118)	0.314
miR-519b-3p	0.556 (0.393–0.787)	<0.001 *	0.605 (0.403–0.908)	0.015 *
miR-519c-3p	0.776 (0.625–0.962)	0.021 *	0.681 (0.499–0.928)	0.015 *
miR-520a-5p	1.009 (0.783–1.300)	0.943	0.944 (0.694–1.284)	0.714
miR-521	0.664 (0.507–0.869)	0.003 *	0.664 (0.478–0.921)	0.014 *
miR-523-3p	0.697 (0.520–0.933)	0.015 *	0.648 (0.441–0.951)	0.027 *
miR-524-3p	0.916 (0.729–1.152)	0.454	0.870 (0.646–1.173)	0.362
miR-524-5p	0.845 (0.623–1.145)	0.277	0.884 (0.604–1.295)	0.528
miR-525-3p	0.816 (0.672–0.990)	0.039 *	0.829 (0.657–1.046)	0.114
miR-525-5p	0.856 (0.697–1.051)	0.138	0.886 (0.688–1.141)	0.347

^1^ Adjusted for maternal age, BMI classification, final gestational weight gain, height, smoking status, polycystic ovary syndrome history, treatment type, physical activity level, gestational age at delivery, newborn sex, maternal glucose and insulin concentrations, and GDM status. * *p* < 0.05. Women with GDM who delivered a full-term newborn with normal birthweight were used as the reference group.

## Data Availability

The datasets and analytical code used in this study are available from the corresponding author upon reasonable request.

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
