# Peer review of "Association of Circulating miRNAs from the C19MC Cluster and IGF System with Macrosomia in Women with Gestational Diabetes Mellitus"

_ijms, 2025, doi:10.3390/ijms26178367_

Round 1
Reviewer 1 Report
Comments and Suggestions for Authors
I have reviewed the work entitled "Association of circulating miRNAs from the C19MC cluster and 2 IGF system with macrosomia in women with gestational diabetes mellitus."
The topic proposed by the authors is interesting. However, some issues in the methodology and results section prevent the paper from being accepted for publication.
Three groups of women with full-term pregnancies were included: a control group without gestational diabetes and a normal-weight newborn; a second group with gestational diabetes and a normal-weight newborn; and a third group of patients with gestational diabetes (GDM) and a large-for-gestational-age (LGA) newborn. The plasma expression of 20 miRNAs from the C19MC cluster was determined by RT-qPCR, and the plasma concentration of 10 IGF axis proteins by ELISA. Insulin, HOMA-IR, and glucose levels are also reported. According to the results reported in Figure 1, they found four miRNAs with statistically significant differences compared to the control group: two (518d-3p and 521) were underexpressed in the GDM and LGA groups, one (516a-5p) was overexpressed only in the GDM group with normal-weight newborns, and one (525-3p) was overexpressed in both GDM groups compared to the control group. Regarding the IGF axis proteins analyzed, only IGF-2 had significantly higher expression in GDM and LGA compared to the control group.
The authors should show, perhaps in a supplementary material section, the raw data from their experiments, including the RT-qPCR graphs and CT values, because they are reporting the relative expression of miRNAs, normalizing with an exogenous miRNA and the miRNA expression of the control group. However, the fold expression values ​​reported in Figure 1 show two miRNAs with >300-fold expression (516a-5p, 525-3p), one with >2000-fold expression (518f-5p), another with >40,000-fold expression (519c-3p), and another with >150,000-fold expression (523-3p). How do authors explain this expression level from a biological and pathophysiological perspective? How do you explain that, despite these expression levels, not all have a statistically significant difference compared to the control group?
Although the only factor with statistically significant differences in the GDM+LGA group compared to the control group was IGF-2, none of the miRNAs analyzed significantly correlated with this factor, according to the analysis shown.
The results in the logistic regression model are inconsistent.
The statements in the discussion and conclusions section are not supported by the results shown.
Reviewer 2 Report
Comments and Suggestions for Authors
The manuscript entitled “Association of circulating miRNAs from the C19MC cluster and IGF system with macrosomia in women with gestational diabetes mellitus” focuses on the potential biomarker properties of C19MC microRNAs in GDM and the associated fetal macrosomia. The topic of the manuscript is within the scope of the journal, would be interesting to the readership, especially to the clinicians and researchers interested in the molecular basis of GDM and clinical management of diabetic states, provided that the study design is adequate for reaching the intended study aim.
The manuscript is relatively well written and well structured. Material and methods presented most of the required details, while the presentation of the Results is mostly informative. However, the major conclusion about the microRNAs’ “potential as EARLY biomarkers for macrosomia risk in GDM” is exaggerated and not supported by the findings. The design of the study, which is based on the analysis of microRNA concentration in plasma samples derived from blood samples on the day of delivery, does not correspond with the qualification of the analyzed microRNAs as early biomarkers of any GDM-related state. Therefore, interpretation should be restricted to peripartal period. There are also some other major and minor issues that should be resolved and some corrections are needed:
- Taking blood from patients before delivery could result in a bias, since the onset of a spontaneous labor can result in physiological changes and microRNAs release which could affect the results of circulatory RNA quantification. For this reason, only patients referred to elective CSs or induced labor are usually selected for sampling for this type of studies. Furthermore, there is a difference in the distribution of delivery modes between study groups.
- Relative expression of microRNAs (2-dCt) is usually a parameter with non-normal distribution. Therefore, mean and SD are not adequate statistical measures of central tendency and data dispersion, while the comparison of data requires a usage of a non-parametric test, or logarithmic transformation, such as using -dCT. The major finding presented in Figure 1 indicate that there are some differences between some of the groups, but there is no obvious trend from group 1 to 3. Additionally, there are no data on the number of samples with negative microRNA quantification results for each microRNA per group.
- If the purpose of this manuscript was to analyze the biomarker potential in terms of pregnancy outcome in GDM, group 1 should not have been used as a reference. There is no data on which group was used as reference for Table 2.
- Subsection 2.4. should be rewritten, since the descriptive analysis of the data presented in mislabeled “Figure 1” (correlation heatmaps) does not indicate which group of patients the results refer to. Also, the correlation results for group 1 are not necessary for reaching the intended study aim. The interpretation of this figure is also inadequate, since the correlations between the microRNA expression and IGF axis biomarkers should be determined in a group of consecutively recruited patients with GDM, regardless of their “LGA status”.
- Line 48: remove “abnormal”, since “intolerance” indicates abnormality per se.
- Lines 87-88: The sentence should be rephrased, since it seems like there is a part missing.
- Line 138: remove repeated text.
Reviewer 3 Report
Comments and Suggestions for Authors
In this study the authors examine the C19MC miRNA expression patterns, IGF family serum concentrations, and their relationship with birthweight in Mexican women with GDM. The paper is interesting, but I have relevant methodological concerns that must be addressed by authors.
Previous studies have demonstrated that the early treatment of GDM is critical for preventing macrosomia (doi: 10.1016/S2213-8587(20)30189-3 and doi: 10.1155/2020/5393952). This effect could be due to the duration of exposition to the hyperglycemia. Neverthless, in the present paper no information is provided about GDM therapy, glucose monitoring, and glycemic results.
Also, the authors should perform a correlation/association analysis between glycemic values at OGTT and during pregnancy with the C19MC miRNA expression patterns and IGF family serum concentrations.
In addition, precise if the enrolled women are consecutive.
Round 2
Reviewer 1 Report
Comments and Suggestions for Authors
I did not accept this paper with major corrections. I reject it because of the lack of transparency and accuracy in the conducted experiments. Authors must provide the raw RT-qPCR experimental data. There were many methodological errors. Authors reassessed the entire statistical analysis, yielding completely different results when comparing with the first version of the manuscript. Yet, the conclusions are pretty much the same. Also, I suggest adhering to MIQE guidelines for reporting RT qPCR results.
Reviewer 2 Report
Comments and Suggestions for Authors
The authors have made corrections according to suggestions and the manuscript is considerably improved. I agree with their choice of including healthy normoglycemic controls in their study. However, I strongly disagree with using this group as a reference for analyzing association with LGA, since any finding could merely reflect the association with GDM, not with LGA characteristic per se. Therefore, GDM with normal weight neonatal outcome should be the reference for this part of the study, regardless of consistency in using control groups, since the purpose is different than in other parts of the research.
Reviewer 3 Report
Comments and Suggestions for Authors
The manuscript has been improved.
Round 3
Reviewer 2 Report
Comments and Suggestions for Authors
The authors have made significant improvements by including corrections according to suggestions.